# Site-Specific Antibody Conjugation to Engineered Double Cysteine Residues

**DOI:** 10.3390/ph14070672

**Published:** 2021-07-14

**Authors:** Qun Zhou, Josephine Kyazike, Ekaterina Boudanova, Michael Drzyzga, Denise Honey, Robert Cost, Lihui Hou, Francis Duffieux, Marie-Priscille Brun, Anna Park, Huawei Qiu

**Affiliations:** 1Large Molecules Research, Sanofi, Framingham, MA 01701, USA; Josephine.Kyazike@sanofi.com (J.K.); Ekaterina.Boudanova@sanofi.com (E.B.); mgdrzyzga@charter.net (M.D.); Denise.Honey@sanofi.com (D.H.); Robert.Cost@sanofi.com (R.C.); Lihui.Hou@sanofi.com (L.H.); Anna.Park@sanofi.com (A.P.); huaweiqiu@yahoo.com (H.Q.); 2Large Molecules Research, Sanofi, 94400 Vitry-Sur-Seine, France; Francis.Duffieux@sanofi.com; 3Integrated Drug Discovery, Sanofi, 94400 Vitry-Sur-Seine, France; Marie-Priscille.Brun@sanofi.com

**Keywords:** site-specific antibody-drug conjugation, THIOMAB^TM^, engineered double cysteine, PEGylation, conjugation efficiency and selectivity

## Abstract

Site-specific antibody conjugations generate homogeneous antibody-drug conjugates with high therapeutic index. However, there are limited examples for producing the site-specific conjugates with a drug-to-antibody ratio (DAR) greater than two, especially using engineered cysteines. Based on available Fc structures, we designed and introduced free cysteine residues into various antibody CH2 and CH3 regions to explore and expand this technology. The mutants were generated using site-directed mutagenesis with good yield and properties. Conjugation efficiency and selectivity were screened using PEGylation. The top single cysteine mutants were then selected and combined as double cysteine mutants for expression and further investigation. Thirty-six out of thirty-eight double cysteine mutants display comparable expression with low aggregation similar to the wild-type antibody. PEGylation screening identified seventeen double cysteine mutants with good conjugatability and high selectivity. PEGylation was demonstrated to be a valuable and efficient approach for quickly screening mutants for high selectivity as well as conjugation efficiency. Our work demonstrated the feasibility of generating antibody conjugates with a DAR greater than 3.4 and high site-selectivity using THIOMAB^TM^ method. The top single or double cysteine mutants identified can potentially be applied to site-specific antibody conjugation of cytotoxin or other therapeutic agents as a next generation conjugation strategy.

## 1. Introduction

Antibody-drug conjugation has been applied extensively in clinics against cancer [1]. There are ten approved antibody-drug conjugates (ADCs) which show promising clinical results for different cancer indications [2,3,4,5,6]. Currently, many ADCs are under clinical development [7,8]. Antibody conjugation can also be utilized in coupling antibodies with other small molecules in addition to cytotoxins, such as antibiotics and PROTAC (Proteolysis-Targeting Chimera) for intracellular protein degradation directed by small molecules [9,10,11,12]. Antibody conjugation combines the advantage of specificity associated with antibodies and high potency of small molecules. Considering the importance of this format and the heterogeneity associated with the first-generation ADC methods, there is a clear need for next generation site-specific conjugation to produce homogeneous conjugates for ease of characterization and reduced adverse effects [13,14,15]. THIOMAB^TM^, one of the first site-specific antibody-drug conjugation approaches developed, is based on engineering to introduce unpaired cysteine residues in specific locations in an antibody molecule for site-specific conjugation [16,17]. It shows not only high homogeneity but also increased efficacy and therapeutic index in vivo in animal models. There are many sites in the antibody Fab and Fc regions that have been engineered to introduce single unpaired cysteine residues for site-specific conjugation using the THIOMAB^TM^ approach [18,19,20,21,22,23]. However, the engineering and conjugation of multiple unpaired cysteines in the antibody Fc region have not yet been comprehensively investigated, and there are limited reports related to the introduction of double or triple cysteines for site-specific antibody conjugation aiming for a drug-to-antibody ratio (DAR) greater than two. Although highly-potent cytotoxins allow ADCs with a DAR of two, higher DAR conjugates may broaden the range of the efficacy towards cancer cells expressing low level of tumor-specific antigens [8]. There are also needs for antibody conjugates with high DAR when they are coupled with payloads other than cytotoxins [24,25,26].

In this work, we investigate the feasibility of engineering multiple unpaired cysteines in the antibody CH2 and CH3 region for site-specific conjugation. The different single cysteine mutants, which have been expressed and characterized, were screened for conjugatability using the THIOMABTM approach. The top mutants containing a single unpaired cysteine were then selected and combined with each other as double cysteine mutants for further conjugation screenings. These top single cysteine mutations were also combined with A118C from the CH1 region, which has been shown to generate site-specific ADCs with increased therapeutic index [16]. The top double cysteine mutants have been identified with high conjugation efficiency and selectivity. We show the site-specific conjugation of double cysteine mutants with DAR of ~4 with low off-site coupling. Our results provide a case study using PEGylation screenings for rapidly identifying different single or double cysteine mutants with optimal properties. These cysteine mutations would allow for the generation of unique sites in antibodies for effective site-specific conjugation.

## 2. Results

### 2.1. Design of Single Unpaired Cysteine Mutants

Twenty-seven sites in the Fc region of IgG1 were selected for substitution with a single unpaired cysteine for site-specific antibody conjugation based on their solvent accessibility from the reported crystal structure and predicted reactivity with thiol-specific conjugation chemistries (Figure 1) [27]. These residues are exposed and located in the loop, α-helix, or β-sheet in the CH2 or CH3 regions as well as the residues around N297, the conserved N-glycosylation site in the C′E loop (Table 1). The structure (PDB 1E4K) of IgG1 Fc in complex with FcγRIII was used for identifying sites for conjugation with minimal impact on FcγR interaction although the receptor is not shown in the figure. Twenty-one out of twenty-seven sites are present throughout the IgG subclasses. Of the six sites that are not fully conserved, four residues are identical between IgG1 and IgG4, which are the most commonly used IgG subclasses for monoclonal antibody therapy, suggesting the generality of these twenty-seven sites for potential antibody conjugation.

### 2.2. Expression and Characterization of Engineered Single Cysteine Mutants

In order to generate novel unpaired cysteine mutants for conjugation screening, twenty-seven residues in the CH2 and CH3 region have been selected using structure-guided design. Each of the selected amino acid residues were converted to cysteine using site-directed mutagenesis. The mutants and wild-type antibody were expressed from Expi293 cells. Most of the mutants show comparable concentrations in media to the wild-type antibody except for A339C and D413C (Table 1). The antibodies were purified using protein A columns. They were analyzed using SDS-PAGE under non-reducing condition (Appendix A). Although most mutants show high purity, there are half-antibody species, which are significantly present in five of them: N297C, S298C, R301C, E380C, and T437C. Protein aggregates were also detected, mainly, in five mutants: N384C, G385C, Q386C, Q418C, and V422C. Thermal stability was determined using nanoDSF. There are comparable initial melting temperatures (Tm1) among all mutants except N297C, T299C, Y300C, R301C, K326, and E380C which show lower thermal transition temperatures than the wild-type antibody (Table 1).

### 2.3. PEGylation Screening of Single Cysteine Mutants

The conjugatability of these mutants was investigated using PEGylation. Since the engineered cysteines were frequently found disulfide-bonded with free cysteine or glutathione (GSH) in culture media during expression, dithiothreitol (DTT) was first used to uncap these residues so the thiol group is available for conjugation [11,16,19]. After the antibody mutants were partially reduced with DTT, they were re-oxidized with dehydroascorbic acid (dHAA) to restore the native disulfide bonds. The antibodies were then PEGylated and subsequently analyzed using SDS-PAGE stained with Coomassie blue (Figure 2). The PEGylation of the samples was confirmed by running SDS-PAGE under reducing conditions followed by PEG staining (Appendix A). Interestingly, there is low protein aggregation found in most of the conjugates as shown by non-reducing SDS-PAGE, although at least five conjugates displayed high aggregation before conjugation as shown previously (Figure 2 and Appendix A and Table 1). Significant amounts of half-antibody conjugate species were detected with mutants N297C and S298C under non-reducing SDS-PAGE (Figure 2). This was confirmed by PEG staining (Appendix A).

The mono-PEGylated, un-PEGylated, and multi-PEGylated protein bands were detected on the reducing gels (Figure 2). The percentage of each species was then determined. Different single cysteine mutants were PEGylated at different efficiencies (PAR or PEG-to-antibody ratio) and different selectivities, which were calculated by subtracting both % un-PEGylated and % multi-PEGylated bands (above two PEG conjugated per antibody heavy chain; un-desired species) from % mono-PEGylated band (desired species) (Figure 3). The top ten single cysteine mutants were selected from these twenty-seven mutants based on their high conjugation efficiency (PAR ≥ 1.7) and selectivity (≥60% mono-PEGylated, <20% un-PEGylated, and ≤20% multi-PEGylated): A339C, S440C, K290C, S442C, K274C, V422C, N384C, G385C, Q418C, and K360C. Another eleven mutants also showed PAR greater than 1.7, but they were excluded from the top ten list due to their low selectivity, the presence of significant amounts of half antibody conjugates, or low thermal stabilities (Table 1).

In order to further understand conjugation stability, we investigated the molecular integrity of PEG bonds in the conjugates prepared using the top single cysteine mutants after an incubation in plasma as described [30]. The PEGylated single cysteine mutants, including A339C, S440C, S442C, K274C, V422C, N384C, and G385C, were incubated with mouse plasma for 96 h They were analyzed using western blot with an anti-PEG antibody. The PEGylated mutants display high stability with at least 75% of PEGylation remaining on the protein after incubation, when compared to samples without incubation (Figure 4 and Appendix A).

The conjugation using DTT and tris(2-carboxyethyl)phosphine hydrochloride (TCEP) for partial reduction was also compared (Figure 5). In general, the PEGylation using DTT reduction generates conjugates with higher PAR than those using TCEP reduction. Interestingly, different selectivity was observed among the mutants when either DTT or TCEP was used. PEGylation using DTT reduction resulted in better selectivity than TCEP reduction for at least nine single cysteine mutants, including K290C, N297C, Q418C, N384C, S422C, T437C, V422C, S440C, and K360C. In contrast, DTT reduction resulted in poor selectivity in comparison with TCEP reduction for five mutants, including K326C, T299C, Y300C, K414C, and R301C. Four single cysteine mutants, A339C, K274C, G385C, and S440C, showed good conjugation efficiency and selectivity regardless of which reducing agent was used. These four mutants were selected in the top ten conjugation lists from the conjugations with either DTT or TCEP as reducing agent. Two mutants, S383C and A431C, failed to display significant conjugation efficiency and selectivity when either method was applied.

Since DTT reduction followed by PEGylation generated conjugates with higher PAR, it was selected for partial reduction used in conjugation screening of subsequent mutants.

### 2.4. Expression and Characterization of Engineered Double Cysteine Mutants

Thirty-eight double cysteine mutants were designed based on the results of PEGylation screening of single cysteine mutants. The top ten single cysteine mutations from PEGylation with DTT reduction (with the exception of Q418C) were combined with each other or with the previously reported A118C [16]. They were generated using site-directed mutagenesis and expressed from Expi293 cells. All but one of the engineered double cysteine mutants show comparable expression. The exception, K360C + K290C, shows at least 4-fold reduced concentration in media (Table 2). SDS-PAGE showed high levels of aggregation for the mutant S440C + N384C (Appendix A), while SEC-HPLC analysis also detected high levels of aggregates for this double cysteine mutant, as well as A339C + S440C. The thermal stability of these mutants was also investigated. For six of the eight A339C-containing double cysteine mutants for which Tm1 could be measured, there were reduced thermal transition temperatures (Tm1 reduction ≥ 2 degrees) as compared to the wild-type antibody (Table 2).

### 2.5. PEGylation Screening of Double Cysteine Mutants

The double cysteine mutants were screened for conjugation efficiency and selectivity using PEGylation. Different amounts of DTT, dHAA, and PEG were initially investigated for optimal conditions before the PEGylation procedure was applied for screening (data not shown). The mutants were partially reduced using DTT to uncap the engineered double cysteine residues. After re-oxidation with dHAA, the antibody mutants were conjugated with PEG. The PEGylated mutants were analyzed using SDS-PAGE and stained with Coomassie blue (Figure 6). Most of the mutants showed low aggregation after PEGylation when analyzed using SDS-PAGE under non-reducing conditions. In addition, a double cysteine mutant, S442C + V422C, shows no conjugation comparable to the wild-type antibody being tested. The PEGylation of this mutant was repeated with similar results. As shown in Figure 7, the double cysteine mutants show variable conjugation efficiency (PAR). The conjugation selectivity was calculated by subtracting both % un-PEGylated and % multi-PEGylated (greater than three PEG conjugated per antibody heavy chain; undesired species) from % mono- and di-PEGylated antibodies (desired species), and it was found variable among the mutants. Out of thirty-eight mutants, seventeen mutants display a PAR greater than 3.4 after conjugation. They also show good selectivity (≥70%) with mono- and di-PEGylated species above 80%, un-PEGylated species below 5%, and multi-PEGylated species below 10% (Table 3 and Figure 8). The top double cysteine mutants include A118C paired with A339C, G385C, K274C, N384C, S440C, or V422C; K274C paired with A339C, G385C, N384C, S440C, or V422C; K290C paired with N384C; and A339C paired with G385C, K290C, N384C, S440C, or V422C.

The top double cysteine mutants demonstrated not only high coupling efficiency but also good selectivity. The A118C, A339C, and K274C mutations are highly compatible with other cysteine mutants including each other. The PEGylation of large numbers of cysteine-containing mutants allows for simple and straightforward screening to identify top antibody mutants with a PAR greater than 3.4 and minimal off-target conjugation.

## 3. Discussion

Site-specific antibody conjugation generates homogeneous conjugates, simplifies characterization, and increases therapeutic index. Many different methods have been developed for ADCs with a DAR of two, which is sufficient for highly-potent cytotoxins [16,19,31,32,33,34,35,36,37,38,39,40,41,42,43,44]. However, there are needs for site-specific antibody conjugates with a DAR greater than two when other payloads besides cytotoxins are coupled. Currently, examples of site-specific conjugation for ADCs with a DAR greater than two are limited, and there is no information related to conjugation selectivity available [24,45]. In this work, we engineered twenty-seven single and thirty-eight double cysteine residues in the Fc region using site-directed mutagenesis. The conjugatability of these single and double cysteine mutants has been investigated using PEGylation to generate conjugates with a DAR around three to four. Our results suggest that at least seventeen double cysteine mutations in the CH2 and CH3 regions can be engineered for site-specific conjugation using the THIOMAB^TM^ approach. Seventeen double cysteine mutants have been identified for high conjugation efficiency and selectivity for the conjugates with a DAR greater than 3.4. Our method using PEGylation screening is a simple and robust process which can potentially be applied to screening additional mutants.

Different sites in the antibody CH2 and CH3 regions were selected for cysteine substitution based on solvent accessibility from crystal structures [27]. Although most of the mutants were expressed at comparable levels to the wild-type antibody, a few mutants showed significant half-antibody species in SDS-PAGE under non-reducing conditions. These mutants include those with mutations around the conserved N297 site, including N297C and S298C. It is likely that these two free cysteine residues result in disulfide scrambling since they are close to C229 at the low hinge region based on IgG1 Fc structure, which is involved in an interchain disulfide bond formation. The disruption of the interchain disulfide can potentially lead to half antibody formation, which was still present after PEGylation. In addition, other factors cannot be ruled out for potentially contributing to the generation of half antibody species, such as conformational change.

There are a few reports on conjugation through engineered single cysteines in the CH2 and CH3 region. Jeffrey et al. showed site-specific conjugation of the pyrrolobenzodiazepine dimer in S239C which displays strong in vitro and in vivo antitumor activity [19]. A single cysteine mutant, S442C, was also generated in IgG4 and shows efficient conjugation with a bifunctional chelator, bromoacetyl-TMT [20]. Additional engineered single cysteine mutants have been reported for labeling, including K290C, K326C, K334C, Q347C, S375C, E380C, E388C, K392C, S415C, S440C, N421C, and L443C in the CH2 and CH3 regions [21,22]. Among these previously reported mutants, seven single cysteine mutants, including S239C, K290C, K326C, E380C, S415C, S440C, and S442C, have been evaluated in this work. Among them, only three mutants, K290C, S440C, and S442C, were on our top ten list derived from the screening of single cysteine mutants. The reason why other mutants, including S239C, K326C, E380C, and S415C, do not perform well in conjugation could be due to the difference in selection criteria and the method of detection. Our PEGylation screening selects the top mutants not only based on good conjugation efficiency but also high selectivity. In this study, three single cysteine mutants, K326C, E380C and S415C, show high multi-PEGylation and un-PEGylation species while S239C displays low mono-PEGylation and high un-PEGylation compared to the top ten mutants. Thus, their conjugation selectivities do not satisfy our stringent selection criteria.

Antibody conjugation efficiency, such as DAR or PAR, is an important criterion for assessing the success of a coupling reaction. Moreover, selectivity is also a relevant parameter for comparing the different conjugates. Behrens et al. reported an efficient site-specific conjugation of the antibody using a thiol bridge method with a bifunctional dibromomaleimide linker for cytotoxin coupling through cysteines in hinge disulfides [46]. Although their ADC displays improved PK and reduced toxicity in vivo compared to analogous conventional cysteine ADCs, ~70% of conjugates crosslinked both cysteines from interchain disulfides and ~30% resulted in half antibody conjugates as assessed by denaturing SEC analysis. Since the partial reduction and re-oxidation steps are used in the THIOMAB^TM^ approach, there is a possibility of introducing disulfide scrambling or incomplete re-oxidation, which could contribute to multiple conjugated species for several suboptimal mutants through either under-oxidized cysteine or coupling to other amino acid residues [12]. The high amounts of multiple conjugated species, which result from off targeted site conjugation, can display higher DAR or PAR with low selectivity, leading to un-desired heterogeneity.

It is known that maleimide reactions generate a thiosuccinimide group which is susceptible to the retro-Michael reaction, leading to a loss of payloads from the antibody into other proteins such as albumin [22,30]. The drug-linker is rapidly lost from a mutant with a highly solvent-accessible site, Fc-S396C, by reacting with free thiols in albumin, free cysteine, or glutathione through maleimide exchange [30]. In contrast, the drug-linker from an ADC containing a partially accessible site, LC-V205C, with a positively charged environment is stable in plasma. While the linker at the site, HC-A118C, with partial solvent-accessibility and neutral charge shows intermediate stability. The plasma stability of our top single cysteine mutants after PEGylation was also investigated. There are less than 30% increases of three mono-PEGylated conjugates, including A339C-PEG, S440C-PEG, and N384C-PEG, from time 0 to 96 h, probably due to variations of western blotting analysis, which is known to be semi-quantitative. Nevertheless, all the conjugates prepared using top single cysteine mutants seem to be relatively stable in mouse plasma after incubation at 37 °C for four days. Our results are consistent with a report showing similar plasma stability of ADCs prepared using mutants containing engineered single cysteine residues in the Fc region [22].

Compared to other methods reported for screening mutants for site-specific conjugation [18,37,47,48,49,50], the current work provides an alternative approach for rapid screening using PEGylation. Our method is also simple and straightforward. It provides information not only related to conjugation efficiency, but also about the selectivity of each mutant. Therefore, an additional stringent criterion can be applied to identify the top mutants and to select optimal sites for site-specific conjugation. Although PEG is relatively large and more hydrophilic than fluorescent dyes, PEGylation screening provides more useful information than a screening that uses fluorescent dyes, which are often small in size and provide no information about selectivity. Moreover, PEGylation allows simple characterization using SDS-PAGE without purification, eliminating numerous steps, and reducing variables in the analysis. It also involves no fluorescence quenching or differences in ionization, yielding more consistent results than other methods.

In this study, the partial reduction in uncapping engineered cysteines using DTT and TCEP was also compared during PEGylation screening of single cysteine mutants since both reagents have been extensively used in cysteine conjugation [10,11,16,19]. Lower conjugation efficiency was observed with PEGylation using TCEP than using DTT reduction. Interestingly, there is a significant difference among mutants in conjugation selectivity. At least nine single cysteine mutants PEGylated with DTT show better selectivity, while five others display poorer selectivity than those using TCEP reduction. The difference could be due to the variation in local hydrophobicity or electrostatic profiles around the mutation sites as these two reducing agents have different chemical properties, such as charge. TCEP reduction can potentially be applied to mutants which do not show good conjugation selectivity using DTT.

A variety of site-specific antibody conjugation methods have been developed for increased homogeneity and therapeutic index of ADCs, including those using unnatural amino acids, engineered amino acids, microbial transglutaminase, and glycans. Among those methods, many of them utilize a single site for engineering, leading to a DAR of ~2. With different therapeutic applications or the desire for high therapeutic index, it would be beneficial to have various conjugation options such as methods for generating antibody conjugates with a DAR greater than two with different payloads. With this in mind, we used PEGylation screening to rapidly investigate the feasibility of conjugating engineered single and double cysteine mutants using the THIOMAB^TM^ approach. The PEGylation demonstrated in this study allows us to compare not only the conjugation efficiency but also the selectivity of different mutants. Thus, our top double cysteine mutants can be applied to site-specific antibody conjugation with a DAR of ~4, potentially for better therapeutic needs.

## 4. Methods

### 4.1. Site-Directed Mutagenesis

Episomal expression vector pFF [51], encoding a monoclonal antibody IgG1, was used as a template DNA. Single or double cysteine residues were introduced by using QuikChange Lightning Site-Directed mutagenesis kit (Agilent, Santa Clara, CA, USA, Catalog No. 210519) or Q5 Site-Directed mutagenesis kit (New England BioLabs, Ipswich, MA, USA, Catalog No. E0554S), according to the manufacturer’s instructions. The full coding region was sequenced to confirm the cysteine mutations being introduced.

### 4.2. Transfection and Purification

Expression of single unpaired cysteine mutants was performed in Expi293F suspension cells. The cells were transiently transfected with mutated DNA plasmid using Expi293 expression system (ThermoFisher Scientific, Waltham, MA, USA, Catalog No. A14635) in 96-well plates at 1.25 × 10^6^ cells/0.5 mL/well, according to the manufacturer’s instructions. Each mutant was expressed in multiple wells (6 or 10 wells) to harvest 3 or 5 mL of conditioned media for Protein A purification. The 3 mL harvests were applied to 80 µL PhyNexus tips (San Jose, CA, USA) on a Hamilton Microlab STAR Liquid Handling System (Hamilton, ON, Canada). The proteins were captured, washed, and eluted with proprietary PhyNexus buffers (San Jose, CA, USA). The 5 mL harvests were purified using 1-mL HiTrap protein A columns (Cytiva, Marlborough, MA, USA) on a Protein Maker (Protein BioSolutions, Gaithersburg, MD, USA). Double cysteine mutants were also expressed from Expi293 cells at 10 mL scale in 50 mL bioreactor tubes. They were purified using 1-mL MabSelect Sure columns (Cytiva, Marlborough, MA, USA) on a Protein Maker and buffer-exchanged into PBS using Amicon-15 filters (MWCO 10 kDa).

### 4.3. SDS-PAGE Analysis

Protein samples (3–4 µg) were heated for 10 min at 70 °C in reducing or non-reducing buffer (ThermoFisher Scientific, Waltham, MA, USA), run on 4–12% Bis-Tris NuPAGE (ThermoFisher Scientific, Waltham, MA, USA) or 4–12% Tris-glycine gel (ThermoFisher Scientific, Waltham, MA, USA) and stained with Coomassie blue (Imperial™ Protein Stain, ThermoFisher Scientific, Waltham, MA, USA) or PEG stainings. The mono-PEGylated, un-PEGylated, and multi-PEGylated protein bands were detected for each sample within a single lane on the Coomassie blue stained reducing gels, which were also scanned using ProteinSimple (San Jose, CA, USA). The percentage of each species was determined with AlphaView software from ProteinSimple (San Jose, CA, USA) [29]. Since the results were found consistent and wild-type antibody is included in either conjugation or SDS-PAGE, no replicated analysis is needed for each sample during the characterization.

The PEG staining was run according to a report with modification [52]. 13 µg of PEGylated antibody samples were applied to 4–12% NuPAGE under reducing or non-reducing conditions. After being extensively rinsed in water, the gels were treated with 5% BaCl_2_ for 10 min at room temperature and washed with water. They were then stained with a solution containing 20 mM potassium iodide and 10 mM iodine for ~3 min until PEGylated antibody bands appear.

### 4.4. Plasma Stability Study

PEGylated single cysteine mutants, which have been purified using buffer-exchange with Amicon ultracel filters (MWCO 50kDa), and wild-type antibody were incubated with mouse plasma (Innovative Research) with the concentration of PEGylated antibody at 0.02 mg/mL, which is close to injected dose in vivo, for 0 and 96 h at 37 °C in CO_2_ incubator as described [30]. The samples for 0–h time point were frozen in dry-ice immediately and then kept at −80 °C before analysis. After incubation, 0.1 µg of each sample was applied to 4–12% NuPAGE under reducing condition. The proteins were transferred to PVDF membranes using iBlot 2 Dry Blotting System according to manufacturer’s instruction (ThermoFisher Scientific, Waltham, MA, USA). The membrane was first blocked with SuperBlock blocking buffer in PBS (ThermoFisher Scientific, Waltham, MA, USA). The western blot was performed by incubation of the PVDF membranes with biotinylated anti-PEG rabbit monoclonal antibody (1:1000, from ThermoFisher Scientific, Waltham, MA, USA, RM105, Catalog No. MA5-27978) in 5% (*w/v*) BSA in PBS containing 0.05% Tween-20 overnight at 4 °C and followed by incubating with streptavidin-HRP (1:5000, from Sigma, Saint Louis, MO, USA) for 1 h at room temperature. After washing with PBS containing 0.05% Tween-20, the bound streptavidin-HRP was detected with SuperSignal West Pico PLUS chemiluminescent substrate (ThermoFisher Scientific, Waltham, MA, USA). The signals were determined using FluorChem system and bands were analyzed using AlphaView software (ProteinSimple, San Jose, CA, USA) [29].

### 4.5. SEC-UPLC

Size exclusion chromatography was performed with 5 μg mutant diluted in mobile phase (PBS, pH7.2), injected and separated onto an ACQUITY UPLC Protein BEH200A SEC column on a UPLC system (Waters, Milford, MA, USA) with flow rate at 0.3 mL/minute for 10 min.

### 4.6. Thermal Stability

The unfolding transition temperatures (Tm) of mutants were analyzed by nanoDSF, using the Nanotemper Prometheus NT.48 instrument (München, Germany). Samples (~10 μL at 1 mg/mL in PBS) were loaded by capillary action into high sensitivity grade capillaries, placed on the Prometheus capillary holder and subjected to a temperature ramping of 0.5 °C/minute from 20 °C to 95 °C. The samples were run in triplicate. The Tm (°C) values indicate the structural stability of the samples and were obtained by monitoring the intrinsic fluorescence at the emission wavelengths of 330 nm and 350 nm. To generate an unfolding curve, the ratio of the fluorescence intensities (F350 nm/F330 nm) was plotted vs. temperature or time. The thermal stability of a sample was described by the thermal unfolding transition midpoint Tm (°C), at which half of the protein population is unfolded.

### 4.7. Conjugation

The conjugation was performed using the THIOMAB^TM^ approach [11,16,19]. For PEGylation of single unpaired cysteine mutants partially reduced with dithiothreitol (DTT), the engineered cysteines were uncapped with 64 eq of DTT in PBS containing 5 mM EDTA with a layer of argon at 37 °C for 45 min. After buffer-exchange into 20 mM phosphate containing 2 mM EDTA (pH 6.5) using Amicon ultracel filters (MWCO 50kDa), the antibody samples were re-oxidized with 10 eq of dehydroascorbic acid (dHAA) at room temperature for 3 h. The reduced and re-oxidized samples were then PEGylated with 5 eq of 5 kDa maleimide PEG in 16 mM sodium phosphate (pH 8.0) at 1mg/mL final protein concentration at room temperature for 2 h. For PEGylation of single unpaired cysteine mutants partially reduced with tris(2-carboxyethyl)phosphine hydrochloride (TCEP), the engineered cysteines were uncapped with 10 eq of TCEP in 10mM sodium phosphate buffer containing 2mM EDTA with a layer of argon at 37 °C for 2 h. The antibody samples were buffer-exchanged, re-oxidized, and PEGylated as described above. PEGylation was quantified on different bands within a single lane for each sample on SDS-PAGE. PAR was determined as follow: PAR = 2 × [0 × (% un-PEGylated/100) + 1 × (% mono-PEGylated/100) + 2 × (% di-PEGylated/100) + 3 × (% tri-PEGylated/100)], while % multi-PEGylated = % di-PEGylated + % tri-PEGylated. Selectivity was calculated as follows: Selectivity = % mono-PEGylated − % un-PEGylated − % multi-PEGylated antibody.

In PEGylation screening of double cysteine mutants, the antibody samples were reduced with 120 eq of DTT in PBS containing 5 mM EDTA with a layer of argon at 37 °C for 45 min. After buffer exchange into 20 mM sodium phosphate containing 2 mM EDTA (pH 6.5), the samples were re-oxidized with 30 eq of dHAA at room temperature for 3 h. The samples were then PEGylated with 5 eq of 5 kDa maleimide PEG in 84 mM sodium phosphate buffer (pH 7.0) at 1mg/mL final protein concentration at room temperature for 2 h. The quantitation was performed on different bands within a single lane on SDS-PAGE for each sample. PAR was determined as follow: PAR = 2 × [0 × (% un-PEGylated/100) + 1 × (% mono-PEGylated/100) + 2 × (% di-PEGylated/100) + 3 × (% tri-PEGylated/100) + 4 × (% tetra-PEGylated/100) + 5 × (% penta-PEGylated/100) + 6 × (% hexa-PEGylated/100)]. % mono- and di-PEGylated = % mono-PEGylated + % di-PEGylated, while % multi-PEGylated = % tri-PEGylated + % tetra-PEGylated + % penta-PEGylated + % hexa-PEGylated. Selectivity was calculated as follow: Selectivity = % mono- and di-PEGylated—% un-PEGylated—% multi-PEGylated antibody.

The PEGylation of a few cysteine mutants was repeated with wild-type antibody included as a negative control during conjugation.

## Figures and Tables

**Figure 1 pharmaceuticals-14-00672-f001:**
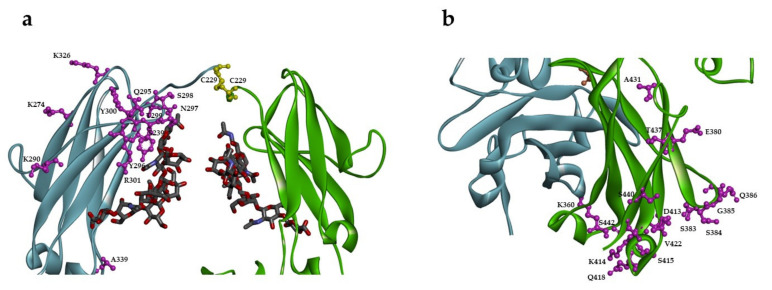
Different sites in Fc region selected for substitution with cysteine for conjugation. (**a**) Sites in the CH2 region with N-glycan attached at N297 (in ball and stick) as shown on one of two heavy chains (left in light blue). The sites are in scaled ball and stick in magenta while two C229 residues in yellow from both heavy chains formed an interchain disulfide bond. (**b**) Sites in the CH3 region as shown on one of two heavy chains (right in green). The sites are in scaled ball and stick in magenta. The Fc structure is from PDB 1E4K.

**Figure 2 pharmaceuticals-14-00672-f002:**
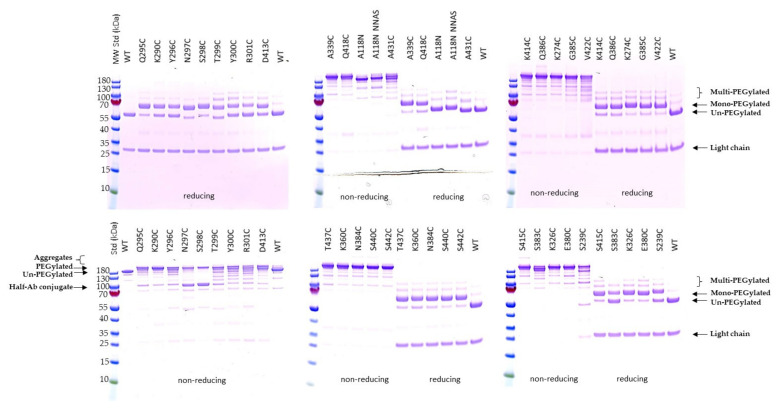
SDS-PAGE analysis of PEGylated single cysteine mutants. 27 unpaired single cysteine mutants were PEGylated and applied to 4–12% Bis-Tris NuPAGE under non-reducing and reducing conditions. The gels were stained with Coomassie blue. The wild-type antibody as well as hyperglycosylated mutants, A118N and A118N NNAS (S298N/T299A/Y300S), were also run together as controls for different migrations of the antibody mutants [28]. PageRuler prestained protein ladder (different kDa as shown on left of the gels) was used as protein molecular weight standards (MW Std or Std). The labels related to identities of the different species are shown (on the left side of non-reducing gel and on the right side of gels under reducing condition).

**Figure 3 pharmaceuticals-14-00672-f003:**
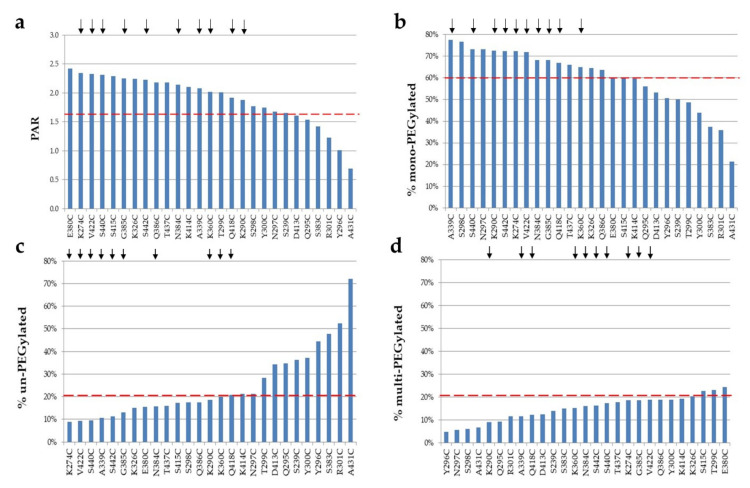
Different conjugation efficiency and selectivity observed among the unpaired single cysteine mutants. The Coomassie blue stained SDS-PAGE reducing gels were scanned using ProteinSimple. The PAR (**a**) and percentages of mono-PEGylated (**b**), un-PEGylated (**c**), and multi-PEGylated (**d**) antibody bands were determined with AlphaView software [29]. The red dot lines represent cut offs, and arrows represent top ten single cysteine mutants.

**Figure 4 pharmaceuticals-14-00672-f004:**
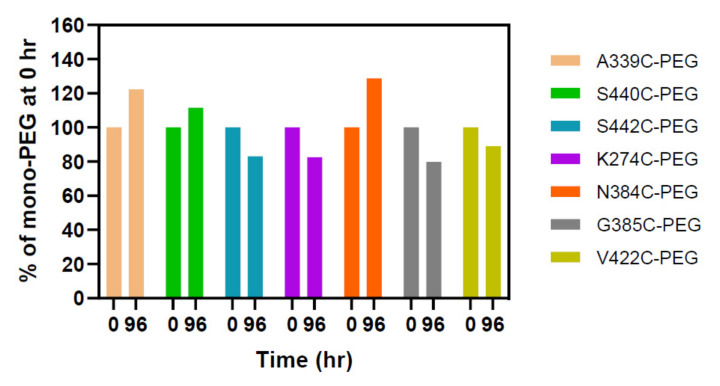
Plasma stability of PEGylated single cysteine mutants. The PEGylated antibodies were incubated with mouse plasma for 0 and 96 h before being analyzed using western blot with anti-PEG antibody. The percent of mono-PEG at 0 h represents the band area of mono-PEGylated species in the sample at 96 h divided by that at time 0, and then multiplied by 100.

**Figure 5 pharmaceuticals-14-00672-f005:**
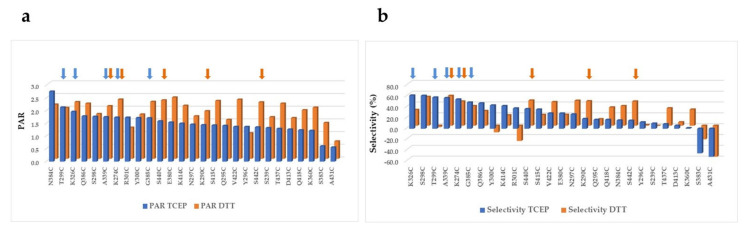
Comparison of conjugation efficiency (**a**) and selectivity (**b**) of single cysteine mutants PEGylated after DTT and TCEP partial reduction. The single unpaired cysteine mutants were partially reduced with either DTT or TCEP to uncap the engineered cysteine residues. After being re-oxidized, the antibody samples were PEGylated and analyzed for conjugation efficiency and selectivity as described in Methods. Selectivity = % mono-PEG% un-PEG—% multi-PEG. The arrows in orange represent the top five mutants identified by conjugation with DTT, while the arrows in blue show the top five mutants identified by conjugation with TCEP.

**Figure 6 pharmaceuticals-14-00672-f006:**
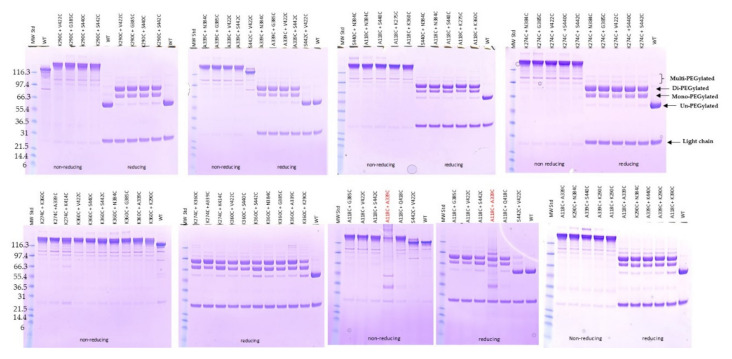
SDS-PAGE analysis of PEGylated double cysteine mutants. 38 unpaired double cysteine mutants were PEGylated and applied to 4–12% Bis-Tris NuPAGE under non-reducing and reducing conditions. The gels were stained with Coomassie blue. The wild-type antibody was also PEGylated and run together as controls. The lanes with protein molecular weight standard are labeled as MW Std with different kDa as shown on left of the gels. The initial clone for mutant, A118C + A339C (highlighted in red), showed poor expression and diffuse bands after PEGylation. It was re-sequenced and found to have a sequence mismatch. A second clone for A118C + A339C with the correct sequence was expressed and PEGylated as shown (labelled in black). The labels related to identities of the different species are shown on the right side of top gel under reducing condition.

**Figure 7 pharmaceuticals-14-00672-f007:**
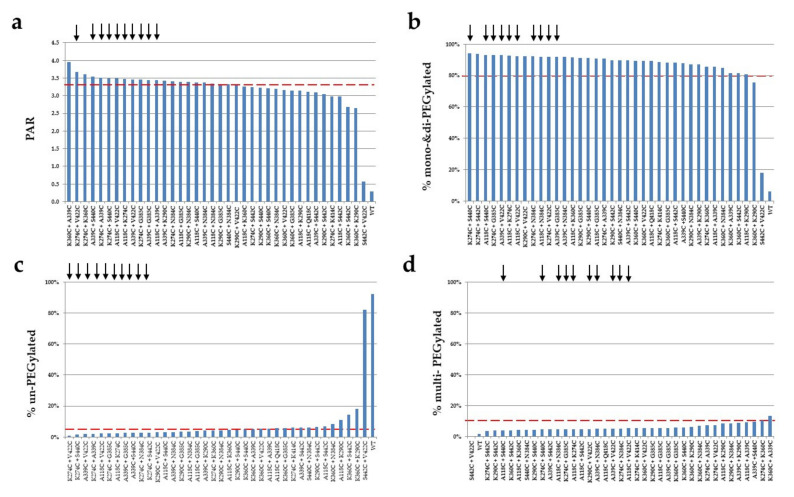
Different conjugation efficiency and selectivity observed among the double cysteine mutants. The double cysteine mutants were analyzed using reducing SDS-PAGE and the gels were stained with Coomassie blue and scanned using ProteinSimple. The PAR (**a**) and percentages of mono- and di-PEGylated (**b**), un-PEGylated (**c**), and multi-PEGylated (**d**) antibody bands were determined with AlphaView software [29]. The red dot lines represent cut offs, and arrows represent top ten double cysteine mutants.

**Figure 8 pharmaceuticals-14-00672-f008:**
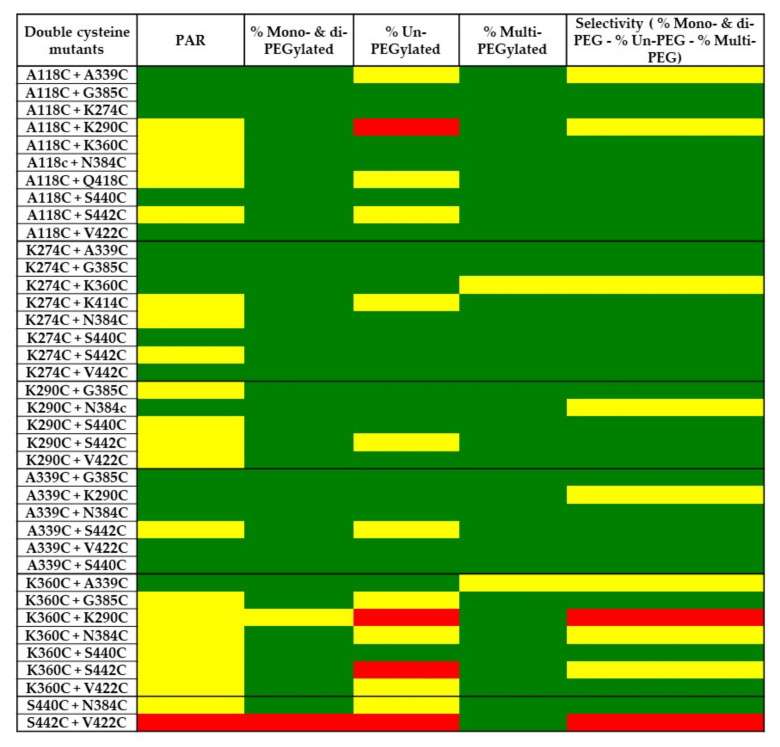
Heatmap of PEGylated double cysteine mutants. Different colors represent ranges of PAR, mono- and di-PEGylated, multi-PEGylated, and un-PEGylated species. PAR: > 3.4 Green < 4; ≥ 2.5 Yellow < 3.4; > 0 Red < 2.5. Mono- and di-PEGylated: > 80% Green < 100%; ≥ 30% Yellow < 80%; > 5% Red < 30%. Un-PEGylated: ≤ 5% Green > 0%; > 5% Yellow ≤ 10%; > 10% Red < 100%. Multi-PEGylated: < 10% Green > 0%; > 10% Yellow < 20%; > 20% Red < 100%. Selectivity (selectivity = % mono- and di-PEG—% un-PEG—% multi-PEG): ≥ 70% Green < 100%, ≥ 60% yellow < 70%, > 0% red < 60%.

**Table 1 pharmaceuticals-14-00672-t001:** Generation and characterization of engineered single cysteine mutants.

mAb	Secondary Structure and Location of Mutation in Fc	mAb Concentration (µg/mL) *	Stability (Tm1, °C) **	Half-Ab ***	Aggregates ***
S239C	Loop, CH2	180, 190	69.8		
K274C	β-sheet, CH2	210	71.9		
K290C	Loop, CH2	130, 250	72.2		
Q295C	Loop, CH2	240, 260	71.2		
Y296C	Loop, CH2	250, 270	70.6		
N297C	Loop, CH2	180, 260	57.6	++	
S298C	Loop, CH2	190, 220	72.0	++	
T299C	Loop, CH2	200, 260	61.2		
Y300C	Loop, CH2	170, 260	62.7		
R301C	Loop, CH2	240	62.3	+	
K326C	Turn, CH2	170	66.5		
A339C	Loop, CH2	100	68.0		
K360C	Loop, CH3	190	70.1		
E380C	β-sheet, CH3	130, 200	63.3	+	
S383C	Loop, CH3	190, 240	68.8		
N384C	Loop, CH3	160, 170	68.7		+
G385C	Loop, CH3	200, 210	69.7		++
Q386C	Loop, CH3	200	70.9		++
D413C	β-sheet, CH3	110	69.9		
K414C	α-helix, CH3	210	71.7		
S415C	α-helix, CH3	160, 160	68.7		
Q418C	α-helix, CH3	180	70.2		+
V422C	Loop, CH3	240	68.3		+
A431C	Loop, CH3	170	67.0		
T437C	β-sheet, CH3	190	67.9	+	
S440C	β-sheet, CH3	150, 150	69.4		
S442C	Loop, CH3	170, 220	68.4		
Wild-type		140, 230	68.8		

Notes: * those with two number represent from two preparations; the mutants with concentration in media below 130 µg/mL are highlighted in red; ** the stability was measured using nanoDSF and mutants with Tm1 reduction ≥2 °C as compared to wild-type are shown in red; *** the results are based on the analysis using non-reducing SDS-PAGE: + indicates the presence of a band for half-antibody or aggregate, while ++ represents that these samples contain the strongest intensity bands for half-antibody or aggregates among all the samples being analyzed (Appendix A).

**Table 2 pharmaceuticals-14-00672-t002:** Characterization of engineered double cysteine mutants.

mAb	mAb Concentration (µg/mL) *	Stability (Tm1, °C) **	% Monomer ***
A118C + N384C	138	67.7	93.3
A118C + G385C	336	68.2	94.9
A118C + V422C	337	67.9	97.4
A118C + S440C	177	69.8	89.2
A118C + S442C	346	68.3	95.7
A118C + K290C	424	ND	NA
A118C + K274C	190	69.3	95.7
A118C + A339C	414	58.8	99.4
A118C + K360C	189	68.3	96.0
A118C + Q418C	395	69.0	93.9
K274C + N384C	300	69.6	87.0
K274C + G385C	248	70.8	84.9
K274C + V422C	499	69.9	94.3
K274C + S440C	361	71.7	85.4
K274C + S442C	346	69.5	94.3
K274C + K360C	381	70.4	97.4
K274C + A339C	296	66.4	97.9
K274C + K414C	332	72.3	96.5
K290C + N384C	146	ND	92.1
K290C + G385C	224	74.1	89.4
K290C + V422C	361	72.9	92.9
K290C + S440C	243	74.2	89.3
K290C + S442C	329	73.3	94.4
A339C + N384C	267	66.0	89.5
A339C + G385C	335	65.0	84.9
A339C + V422C	343	65.9	93.6
A339C + S440C	365	ND	77.0
A339C + S442C	366	65.5	94.0
A339C + K290C	372	ND	96.7
K360C + V422C	320	68.6	97.3
K360C + S440C	476	69.3	90.8
K360C + S442C	323	69.0	97.4
K360C + N384C	335	69.0	89.0
K360C + G385C	330	68.3	90.1
K360C + A339C	375	66.9	98.5
K360C + K290C	33	72.5	98.1
S440C + N384C	308	70.2	45.7
S442C + V422C	345	69.5	97.6
Wild-type	293	68.5	98.1

Notes: * The double cysteine mutant with concentration in media below 130 µg/mL is highlighted in red; ** the stability was measured using nanoDSF and the mutants with Tm1 reduction ≥2 °C as compared to wild-type are shown in red. ND represents not determined; *** the percent monomer was determined using SEC-UPLC and the mutants with monomer species below 85% are in red.

**Table 3 pharmaceuticals-14-00672-t003:** Top double cysteine mutants identified through PEGylation screening *.

Mutant	PAR	% Mono- and Di-PEGylated	% Multi-PEGylated	% Un-PEGylated	Selectivity (% Mono- and Di-PEG—Un-PEG—Multi-PEG)
A118C + A339C	3.4	85.5%	9.0%	5.4%	71%
A118C + G385C	3.4	90.9%	5.4%	3.7%	82%
A118C + K274C	3.5	92.8%	4.7%	2.5%	86%
A118C + N384C	3.4	92.0%	4.6%	3.4%	84%
A118C + S440C	3.4	93.0%	4.0%	3.0%	86%
A118C + V422C	3.5	92.4%	5.3%	2.3%	85%
K274C + A339C	3.5	90.8%	7.2%	2.0%	82%
K274C + G385C	3.5	92.9%	4.6%	2.4%	86%
K274C + N384C	3.4	92.1%	5.1%	2.8%	84%
K274C + S440C	3.5	94.0%	4.5%	1.5%	88%
K274C + V422C	3.7	92.0%	7.2%	0.8%	84%
K290C + N384C	3.4	87.2%	8.5%	4.4%	74%
A339C + G385C	3.4	92.0%	5.5%	2.6%	84%
A339C + K290C	3.4	87.0%	9.0%	4.1%	74%
A339C + N384C	3.4	91.9%	4.9%	3.2%	84%
A339C + V422C	3.5	92.9%	5.1%	2.0%	86%
A339C + S440C	3.5	87.9%	9.5%	2.6%	76%

Notes: * The top mutants were selected according to the criteria: PAR ≥ 3.4, mono- and di-PEGylated ≥ 80%, un-PEGylated ≤ 5%, multi-PEGylated < 10%, and selectivity (selectivity = % mono- and di-PEG—% un-PEG—% multi-PEG) >70%.

## Data Availability

Data is contained within the article and Supplementary Material.

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
