# Peer review of "Site-Specific Antibody Conjugation to Engineered Double Cysteine Residues"

_pharmaceuticals, 2021, doi:10.3390/ph14070672_

Round 1

Reviewer 1 Report

Zhou et al. investigated the feasibility of engineering multiple unpaired cysteine residues in several Fc regions of antibodies. The authors used structure-guided design together with site-directed mutagenesis, conjugation chemistry, plasma, and thermal stability to produce and validate free cysteines in the CH2 and CH3 regions of the Fc domain. Additionally, the author relied on a PEGylation assay as a simple and robust process to screen for the best single cysteine mutants to further combine in double cysteine mutants. Finally, the author demonstrated the feasibility of generating drug-to-antibody ratios of over 3.4 with high selectivity. Although the work lacks mechanistic information and lacks conceptual novelty, it has significance for future optimizations of drug-to-antibody conjugation ratios for preclinical work. Overall, the study was well conducted and of general interest to the readership, however, additional experiments and proper statistics are needed to support their claims, in addition to some work on re-writing the story. Based on this, I recommend publication after the revisions outlined below are fully addressed.

Major points

  1. Conclusions must be based on quantitative and statistically significant data. The author needs to repeat key experiments, add error bars, and appropriate statistics. Also, indicate the number of independent experiments in the figure legends. This applies to all figures and tables in the manuscript.
  2. In page 3, lines 2-13, the authors need a sentence stating the goals of these experiments as this section looks more like a methods section. e.g. “ in order to understand x, we did y, and we found z.
  3. In page 3, line 16. Is there a better term for “capped”? Can the authors re-write this to a more scientific terminology?
  4. In page 4, figure 2, the authors need to add labels and arrows on the left side of gels to indicate what each band is; figure 6 serves as a good example. Please indicate where in the gel the reader can find: aggregation, half-antibody conjugates, mono-un-multi PEGylations.
  5. Related to the point above, these labels should be included in all figures with gels (main and supplementary) throughout the manuscript.
  6. In figure 3, titles in the y-axis are missing, please include. Also, include arrows or starts to indicate the top ten mutations “A339C…..K360C”. This applies to figure 7 as well.
  7. In page 5, line 2, the authors need a sentence stating the goals of these experiments as this section looks more like a methods section. e.g. “ in order to understand x, we did y, and we found z.
  8. In page 6, figure 4, the author needs to explain why the % of mono-PEG increases from time 0 to 96 hrs in some of the columns. Include error bars and statistics.
  9. In page 7, figure 5, use arrows or starts to indicate the top 4-5 mutants. Please, error bars and statistics are needed.
  10. In page 7, table 2, column 2, change “titer” to “concentration”. In the third column, nanoDSF can be removed and added to the table legend. 

Minor points

  1. Values in table 1 and table 2 need to be aligned properly.
  2. In page 1, introduction, line 5, The authors need to explain to the general readership what PROTAC is.
  3. In page 1, introduction, line 11, The authors need to explain in general terms more about the THIOMAB approach, e.g. “approaches developed, based on………, shows”.
  4. In Figure 1, it is hard to visualize the color difference between red, orange, and dark red. The authors need to select a different combination of colors for the readers to easily visualize these residues.
  5. In Table 1 legend, “Tm1 ≥ 2 degrees below” is confusing to the reader. The author needs to re-write this. perhaps “Tm1 reduction ≥ 2 degrees as compared to wild-type”. This applies to all figure legends with similar confusing descriptions.
  6. In table 1, and throughout the manuscript, change “titer” to “concentration”.
  7. There are a couple of typos and grammar corrections needed throughout the manuscript.

Author Response

Author's Response to Comments from Reviewer 1

Zhou et al. investigated the feasibility of engineering multiple unpaired cysteine residues in several Fc regions of antibodies. The authors used structure-guided design together with site-directed mutagenesis, conjugation chemistry, plasma, and thermal stability to produce and validate free cysteines in the CH2 and CH3 regions of the Fc domain. Additionally, the author relied on a PEGylation assay as a simple and robust process to screen for the best single cysteine mutants to further combine in double cysteine mutants. Finally, the author demonstrated the feasibility of generating drug-to-antibody ratios of over 3.4 with high selectivity. Although the work lacks mechanistic information and lacks conceptual novelty, it has significance for future optimizations of drug-to-antibody conjugation ratios for preclinical work. Overall, the study was well conducted and of general interest to the readership, however, additional experiments and proper statistics are needed to support their claims, in addition to some work on re-writing the story. Based on this, I recommend publication after the revisions outlined below are fully addressed.

Response: The authors really appreciate the comments and suggestions from the reviewer.

Major points:

  • Conclusions must be based on quantitative and statistically significant data. The author needs to repeat key experiments, add error bars, and appropriate statistics. Also, indicate the number of independent experiments in the figure legends. This applies to all figures and tables in the manuscript.

Response: We fully understand the point raised by the reviewer. However, we think that the need for statistical analysis depends on which kinds of assays are used. For example, the quantitation and statistical analysis of duplicated or triplicated samples is required for ELISA or ADCC assays but not SEC-HPLC or LC-MS assays as documented in literatures. As we mentioned in our discussion, the PEGylation allows simple characterization using SDS-PAGE without purification of conjugates which eliminates numerous steps, thus reducing variables in the screening analysis. In fact, the quantitation of different bands from each sample has always been done within a single lane in the gels without any potential variation derived from lane to lane comparison. The PEGylation of some mutants was repeated with wild-type antibody included as a negative control during conjugation (see Methods section, highlight in blue).There is no fluorescence quenching or differences in ionization associated with other methods, resulting in consistent results. Based on our experiences, this method is robust and straightforward. Thus, we don’t need to run the PEGylation/SDS-PAGE with each mutant in triplicates. The results are often reproducible, providing unique advantage of this method as compared to other methods.

In addition, the current method is developed for screening purpose. When there is need, the selected top cysteine mutants will need to be applied for scale-up followed by detail characterizations using different methods including those which need statistical analysis as essential requirement.

  • In page 3, lines 2-13, the authors need a sentence stating the goals of these experiments as this

section looks more like a methods section. e.g. “ in order to understand x, we did y, and we found z.  .

Response: The change has been made according to the suggestion.

3) In page 3, line 16. Is there a better term for “capped”? Can the authors re-write this to a more scientific terminology? 

Response: Yes, we agree. The sentence has been re-written (highlighted in blue).

4) In page 4, figure 2, the authors need to add labels and arrows on the left side of gels to indicate what each band is; figure 6 serves as a good example. Please indicate where in the gel the reader can find: aggregation, half-antibody conjugates, mono-un-multi PEGylations.

Response: Changes have been made in figure 2 with all the labels as suggested. They include those for aggregates, half-antibody conjugates, PEGylated and non-PEGylated on the left of non-reducing gel, while the mono-PEGylated, un-PEGylated, multi PEGylated bands derived from heavy chain as well as antibody light chain are labeled on the right side of the gels under reducing conditions.

5) Related to the point above, these labels should be included in all figures with gels (main and supplementary) throughout the manuscript.

Response: Yes, they are included in all relevant figures with gels through the manuscript.

6) In figure 3, titles in the y-axis are missing, please include. Also, include arrows or starts to indicate the top ten mutations “A339C…..K360C”. This applies to figure 7 as well. 

Response: These are good suggestions. We added the titles in the y-axis in both figures 3 and 7. The arrows to indicate top ten mutations have also been included in all the panels of these two figures.

7) In page 5, line 2, the authors need a sentence stating the goals of these experiments as this section looks more like a methods section. e.g. “ in order to understand x, we did y, and we found.

Response: A sentence has been put at the beginning of the paragraph as suggested (highlight in blue).

8) In page 6, figure 4, the author needs to explain why the % of mono-PEG increases from time 0 to 96 hrs in some of the columns. Include error bars and statistics..

Response: A statement was included in Discussion (highlight in blue) “There are less than 30% increases of three mono-PEGylated conjugates, including A339C-PEG, S440C-PEG, and N384C-PEG, from time 0 to 96 hours, probably due to variations of western blotting analysis which is known semi-quantitative. Nevertheless, all the conjugates prepared using top single cysteine mutants seem to be relatively stable in mouse plasma after incubation at 37 °C for four days.”

Since the western blotting analysis is a semi-quantitative assay, it is impossible to include error bars and statistics. When we ran this experiment, an anti-human Fab secondary antibody was tried first for western blot, but it didn’t work. So anti-PEG antibody was applied instead. If the statistics is really required, a different assay, such as ELISA, needs to be developed. A capture antibody for this process needs to be identified from screening for good anti-human Fab secondary antibody and it would take significant amounts of time before we can reach any statistical conclusion.

9) In page 7, figure 5, use arrows or starts to indicate the top 4-5 mutants. Please, error bars and statistics are needed. 

Response: The arrows have been added in figure 5 to indicate the top 4-5 mutants as suggested. However, we wouldn’t be able to include error bars and statistics unfortunately.

10) In page 7, table 2, column 2, change “titer” to “concentration”. In the third column, nanoDSF can be removed and added to the table legend. 

Response: The changes have been made according to the comments from the reviewer.

Minor points:

1) Values in table 1 and table 2 need to be aligned properly. 

Response: Errors are due to formatting for reviewing.

2) In page 1, introduction, line 5, The authors need to explain to the general readership what PROTAC is.

Response: The explanation for PROTAC has been added “(Proteolysis-Targeting Chimera) for intracellular protein degradation directed by small molecules” (highlight in blue).

3) In page 1, introduction, line 11, The authors need to explain in general terms more about the THIOMAB approach, e.g. “approaches developed, based on………, shows”.

Response: Editing has been made to include “THIOMABTM, one of the first site-specific antibody drug conjugation approaches developed, is based on engineering to introduce unpaired cysteine residues in specific locations in antibody molecule for site-specific conjugation” (highlight in blue).

4) In Figure 1, it is hard to visualize the color difference between red, orange, and dark red. The authors need to select a different combination of colors for the readers to easily visualize these residues.

Response: That is good point, and changes have been made according to the comment from the reviewer.

5) In Table 1 legend, “Tm1 ≥ 2 degrees below” is confusing to the reader. The author needs to re-write this. perhaps “Tm1 reduction ≥ 2 degrees as compared to wild-type”. This applies to all figure legends with similar confusing descriptions. 

Response: The phrase “Tm1 > 2 degree below” has been replaced with “Tm1 reduction > 2 °C” according to the suggestion.

6) In table 1, and throughout the manuscript, change “titer” to “concentration”. 

Response: The “titer” has been replaced with “concentration”. Thanks.

7) There are a couple of typos and grammar corrections needed throughout the manuscript.

Response: The typos and grammar have been checked again, and errors found have been corrected.

Reviewer 2 Report

Zhou and co-workers present a systematic study of the suitability of solvent exposed amino acids within the antibody Fc region for use as engineered conjugation sites. Their aim is to identify pairs of amino acids that can be mutated to cysteine without significant impact on the antibody titer, stability and monodispersity, and that, when conjugated via the THIOMAB™ approach, show high levels of site-specific labelling. To achieve this aim, they first screen single mutants using a simple PEG-based methodology, and then combinatorially combine the best performing mutants. A total of 17 novel double mutants were identified from which PEG-antibody ratios greater than 3 could be prepared.

This work provides a comprehensive evaluation of potential engineered conjugation sites in the IgG1 Fc region, and a useful methodology for screening engineered mutants, however, it currently lacks demonstration of applicability within a therapeutic context. This reviewer is hopeful that subsequent work toward this aim is ongoing.

The manuscript requires extensive minor revisions to improve its reproducibility and clarity before it is suitable for publication. These are listed below according to their position within the text.

Throughout the text:

  • Where you refer to the drug/PEG antibody ratio (DAR/PAR), please use ‘greater than’ instead of ‘over’
  • Please be specific rather than using words like ‘some’ and ‘many’

General points:

  • The generality of the mutation sites across antibody types other than IgG1 is not mentioned. How well conserved are these residues? And what proportion of applications use IgG1 compared to other Ig-types?
  • I presume that the general lack of indication of replicates indicates that unless specified otherwise, all data are n = 1? Please clearly state in the Results and Methods whether or not this is indeed the case.
  • Assuming all data are n = 1, it would be helpful to the general reader to have an indication of the likely repeatability of the data presented. Perhaps, for example, multiple measurements are available for wild-type antibody, and can be used to given an indication of likely data ranges.
  • Selectivity as used in Figures 5, 8 and Table 3 is not sufficiently well defined. From the definition given in the column header of Figure 8 and Table 3, I would be expecting a percentage not a fraction? Please review and correct this definition and also include the definition/calculation within the Methods.
  • Tables 1 and 2 - please ensure the tables are formatted consistently throughout – there is a ‘step’ in column positions partway down both tables.
  • Figures 3 and 7 – please format the numbering on the y-axis to 0 significant figures (i.e. 10 not 10.00), remove the % symbol and provide the y-axis with a title (eg. % un-PEGylated)
  • You present data on the thermal stability and monodispersity of the single and double mutants, but do not fully explain whether or how this information led to your selection of (a) single mutants to progress and (b) top double mutants. How does this information relate in importance compared to the PEGylation data?

Abstract:

  • please define the abbreviation DAR upon first use
  • please replace the non-specific phrase ‘most of double cysteine mutants’ with a more specific phrasing such as ‘x out of y double cysteine mutants’

Introduction:

  • The sentences ‘There are many sites in the antibody Fab and several Fc regions being engineered for unpaired cysteine residues for site-specific conjugation using the THIOMABTM approach.18-23 However, the engineering and conjugation of additional unpaired cysteines in antibody Fc region have not yet been thoroughly investigated. There is limited report related to the introduction of double or triple cysteines…’ initially left me confused. In order to better convey that it is the exploration of multiple unpaired cysteines that is lacking from the current literature, I would suggest re-phrasing something like this: ‘There are many sites in the antibody Fab and Fc regions that have been engineered to generate single unpaired cysteine residues for site-specific conjugation using the THIOMABTM approach.18-23 However, the engineering and conjugation of multiple unpaired cysteines in the antibody Fc region has not yet been comprehensively investigated, and there are limited reports related to the introduction of double or triple cysteines…’
  • The rationale behind the combination of novel mutations with A118C is not sufficiently well described, nor is the original literature on A118C cited at this point. Please rectify.
  • The word ‘good’ in the phrase ‘provide a good case study’ is superfluous.

Results

  • Figures 2, 6, S1, S2, S3 and S4 – please indicate the molecular weights of the standards as you have done in Figure S5.
  • Please include a brief rationale for the use of PDB 1E4K to identify suitable mutation sites. There are many more recent, and significantly higher resolution structures (e.g. 5jii amongst others) that could provide a more precise starting point for your analysis.
  • The words residues, amino acids and side chains in the sentence listing the chosen mutation sites and giving their character (polar, charged, nonpolar) are unnecessary.
  • Table 1 – nanoDSF measurements can also give an indication of aggregation – did these measurements agree with the non-reducing SDS-PAGE analysis?
  • Table 1 notes – please re-format ml as mL, degrees as ºC, and provide a key to the symbols + and ++
  • Last sentence before Figure 2: do you mean S298C not N298C?
  • Figure 2 - The hyperglycosylated mutant A118N NNAS and the A118N mutant are not introduced or set in context – please amend the introduction and/or results section to explain why these mutants have been included. Please define the abbreviation NNAS.
  • Figure 2 – please label the molecular weight standards lane with MW Std as specified in the legend.
  • Figure 2 – left hand gels – I think there is a mistake in the labelling – N297Cn should be N297C?
  • Figure 2 – where possible, please apply the same labelling as used on Figure 6 to indicate the identity of the different species.
  • In the phrase ‘Some other mutants showed high PAR…’, please be more specific and replace ‘some’ with a number.
  • Figures 3 and 7 – please indicate the cut offs that you applied with a horizontal line on these bar graphs, and note this in the figure legend.
  • Figure 4 legend – please correct the legend to read The ‘percent of mono-PEG at 0 hr’ represents the band area of mono-PEGylated species in the sample at 96 hours divided by that at time 0, and then multiplied by 100.
  • The sentence ‘Four single cysteine mutants, A339C, K274C, G385C and S440C, showed better conjugation efficiency and selectivity when either reducing agents were used.’ Lacks clarity. I believe what you mean is ‘Four single cysteine mutants, A339C, K274C, G385C and S440C, showed good conjugation efficiency and selectivity regardless of which reducing agent was used.’?
  • Figure 5 – the words PAR and selectivity in the key are superfluous since they are already in the y-axis title.
  • Figure 5B: please move the x-axis labels to the bottom of the graph below all the bars, so that they can be read.
  • The section titled Expression and characterization of engineered double cysteine mutants lacks clarity and conciseness. I suggest rewording thus: Thirty-eight double cysteine mutants were designed based on the results of PEGylation screening of single cysteine mutants. The top ten single cysteine mutations from PEGylation with DTT reduction (with the exception of Q418C) were combined with each other or with the previously reported A118C.16 They were generated using site-directed mutagenesis and expressed from Expi293 cells. All but one of the engineered double cysteine mutants show comparable expression titers. The exception is the double cysteine mutant K360C+K290C, which shows at least 4-fold reduced titer (Table 2). SDS-PAGE showed high levels of aggregation for the mutant S440C+N384C (Figure S5), while SEC-HPLC analysis also detected high levels of aggregates for this double cysteine mutant, as well as A339C+S440C. The thermal stability of these mutants was also investigated. For six of the seven A339C-containing double cysteine mutants for which Tm1 could be measured, this was reduced significantly (ΔTm1 > -2 ºC) compared to wild-type antibody (Table 2).
  • Table 2 – please define NA in the notes, and reformat degrees as ºC
  • The final list of ‘top’ double mutants might be easier to follow if the amino acids were given in order: i.e. A118C paired with x, y, z; K274C paired with x, y, z; K290C paired with x, y, z and A339C paired with x, y, z. Although I appreciate this will then deviate from how you list the amino acids in Table 2. What was the rationale for arranging in the order that you have chosen for Table 2 (and 3, and Figure 8), rather than amino acid residue number?
  • Figure 6. What do you mean by ‘the original mutated clone with the correct sequence’? Do you mean here simply the correct, expected clone A118C + A339C? Perhaps this would be clearer if you were to re-label the incorrect clone with its actual sequence, whatever that turned out to be?
  • Figure 7 – the text on the x- and y-axes is completely illegible as it is too small – please fix this.
  • Figure 8 – the text on the figure is too small to be legible – please fix this.
  • Figure 8 – the definitions for the colouring scheme in the figure legend and the text to the right of the figure do not agree – which is correct? Please remove the incorrect text. The figure legend should be sufficient.
  • Please consider re-phrasing the last paragraph of the results along these lines: ‘The top double cysteine mutants demonstrated not only high coupling efficiency but also good selectivity. The A118C, A339C, and K274C mutations are highly compatible with other cysteine mutants including each other. The PEGylation of large numbers of cysteine-containing mutants allows a simple and straightforward screening to identify antibodies with PAR over 3.4 and with minimal off-target conjugation.’

Discussion

  • Rationals should be rationale
  • Please specify the number of cysteine residues rather than stating ‘we engineered many single and double cysteine residues’ and later ‘Many double cysteine mutants have been identified’
  • Consider replacing ‘more’ with ‘additional’ or even ‘x additional sites’ (where x = the number of novel/additional sites)
  • In your list of mutants that were evaluated in earlier work, you include S239C and S442C, however, these are not mentioned in the preceding sentence to which this sentence refers?
  • In the sentence ‘selectivity is also a relevant specification’, I think parameter may be a better word to use.
  • Instead of ‘less than expected’, use ‘incomplete’ for conciseness.
  • Instead of ‘always be good to’ use ‘be useful/beneficial/helpful to’

Methods

  • Please state which plasmid/vector(s) were used for the mutagenesis and expression or provide a citation for prior work. As currently worded, the mutagenesis and transfection sections are not fully reproducible in another lab.
  • What do you mean by ‘96-well plate format with 0.5 mL culture media at 3- or 5-mL scale’? What was 0.5 mL and what was 3/5 mL? What cell density/cell count did you use? And what method and reagents were used to do the transfections?
  • What recipe/make/supplier of loading buffer did you use for PAGE? And what make/supplier of gels?
  • Please make sure to provide the reference for the AlphaView software throughout the text (i.e. in the figure legends also).
  • Please provide either more details on the PEG staining (time, temperature) or a citation.
  • CO2 should be CO2
  • Please provide details of the supplier etc for the PVDF and details of the western transfer process (wet, semi-dry, fast?)
  • Please provide catalogue numbers for all antibodies
  • Please provide details of the instrument, capillaries, buffers and protein concentrations used for the nanoDSF measurements.

Supplementary information

  • Did you intend for your supplementary to include the main figures as well?
  • Supplementary figure legend S5: SeaBlue Plus 2 should be SeeBlue Plus2
  • Note that in the second instance in the Results where you refer to Figure S4, this should be Figure S5.

Author Response

Author's Response to Comments from Reviewer 2

Zhou and co-workers present a systematic study of the suitability of solvent exposed amino acids within the antibody Fc region for use as engineered conjugation sites. Their aim is to identify pairs of amino acids that can be mutated to cysteine without significant impact on the antibody titer, stability and monodispersity, and that, when conjugated via the THIOMAB™ approach, show high levels of site-specific labelling. To achieve this aim, they first screen single mutants using a simple PEG-based methodology, and then combinatorially combine the best performing mutants. A total of 17 novel double mutants were identified from which PEG-antibody ratios greater than 3 could be prepared.

This work provides a comprehensive evaluation of potential engineered conjugation sites in the IgG1 Fc region, and a useful methodology for screening engineered mutants, however, it currently lacks demonstration of applicability within a therapeutic context. This reviewer is hopeful that subsequent work toward this aim is ongoing.

The manuscript requires extensive minor revisions to improve its reproducibility and clarity before it is suitable for publication. These are listed below according to their position within the text.

Response: We would like to thank the reviewer for the comments, suggestions and helps.

Throughout the text:

  • Where you refer to the drug/PEG antibody ratio (DAR/PAR), please use ‘greater than’ instead of ‘over’.

Please be specific rather than using words like ‘some’ and ‘many’ 

Response: These are excellent suggestions. We have revised the manuscript according to the comments.

General points:

1)  The generality of the mutation sites across antibody types other than IgG1 is not mentioned. How well conserved are these residues? And what proportion of applications use IgG1 compared to other Ig-types?

Response: The generality of the mutation sites has been included in first paragraph of the result section “Twenty-one out of twenty-seven sites are present throughout the IgG subclasses. Even with the rest of six sites, four residues are identical among IgG1 and IgG4 which are mostly used for monoclonal antibody therapy, suggesting the generality of these twenty-seven sites for potential antibody conjugation” (highlight in blue). Since other Ig classes, such as IgM and IgA, haven’t been applied yet for therapeutics in clinics, they were not discussed here. Thanks for the suggestion.

  • I presume that the general lack of indication of replicates indicates that unless specified otherwise, all data are n = 1? Please clearly state in the Results and Methods whether or not this is indeed the case.

Assuming all data are n = 1, it would be helpful to the general reader to have an indication of the likely repeatability of the data presented. Perhaps, for example, multiple measurements are available for wild-type antibody, and can be used to given an indication of likely data ranges.

Response: Yes, n=1 is good enough for data from screening purpose since PEGylation/SDS-PAGE results are quite consistent and reproducible. As suggested by the reviewer, a statement has been included in the Methods section “Episomal expression vector pFF,49 encoding a monoclonal antibody IgG1, was used as a template DNA”.

We fully agree with the reviewer’s opinion, and we have added a few sentences in the Methods or Discussion sections to indicate the likely repeatability of the data presented: “Since the results were found consistent and wild-type antibody is included in either conjugation or SDS-PAGE, no replicated analysis is needed for each sample during the characterization.”” The PEGylation of some cysteine mutants was repeated with wild-type antibody included as a negative control during conjugation.”” Moreover, the PEGylation allows simple characterization using SDS-PAGE without purification which eliminates numerous steps and reduces variables in the screening analysis. There is no fluorescence quenching or differences in ionization associated with other methods, resulting in consistent results.”

We hope that they are better now.

2)  Selectivity as used in Figures 5, 8 and Table 3 is not sufficiently well defined. From the definition given in the column header of Figure 8 and Table 3, I would be expecting a percentage not a fraction? Please review and correct this definition and also include the definition/calculation within the Methods.

Response: Yes, the selectivity should be defined as a percentage, and inaccuracy in definition has been corrected. The definition has been added to the relevant Figure legends including those in Figures 5 and 8, as well as Table 3 and the Methods (highlight in blue).

3)  Tables 1 and 2 - please ensure the tables are formatted consistently throughout – there is a ‘step’ in column positions partway down both tables.

Response: Errors are due to formatting for reviewing.

4)  Figures 3 and 7 – please format the numbering on the y-axis to 0 significant figures (i.e. 10 not 10.00), remove the % symbol and provide the y-axis with a title (eg. % un-PEGylated)

Response: The changes have been made according to the comments from the reviewer.

5)  You present data on the thermal stability and monodispersity of the single and double mutants, but do not fully explain whether or how this information led to your selection of (a) single mutants to progress and (b) top double mutants. How does this information relate in importance compared to the PEGylation data? 

Response: This is a good point. We have mentioned the stability and monodispersity of single mutants in the “PEGylation screening of single cysteine mutants” in the result section “Other eleven mutants also showed PAR greater than 1.7, but they were excluded from the top ten list due to their low selectivity or the presence of significant amounts of half antibody conjugates or low stabilities (Table 1)”(highlight in blue). As for double cysteine mutants, there are reduced thermal stability of A339C paired with other mutations, but we choose not to remove them from our top list. We are also aware that there is not much aggregates in the conjugates prepared from a few of single or double cysteine mutants which show significant amounts of aggregates before conjugation.

Abstract:

1)  please define the abbreviation DAR upon first use 

Response: The abbreviation of DAR has been defined in the abstract (highlight in blue).

2)  please replace the non-specific phrase ‘most of double cysteine mutants’ with a more specific phrasing such as ‘x out of y double cysteine mutants’ 

Response: The changes have been made according to the suggestion from the reviewer.

Introduction:

1)  The sentences ‘There are many sites in the antibody Fab and several Fc regions being engineered for unpaired cysteine residues for site-specific conjugation using the THIOMABTM approach.18-23 However, the engineering and conjugation of additional unpaired cysteines in antibody Fc region have not yet been thoroughly investigated. There is limited report related to the introduction of double or triple cysteines…’ initially left me confused. In order to better convey that it is the exploration of multiple unpaired cysteines that is lacking from the current literature, I would suggest re-phrasing something like this: ‘There are many sites in the antibody Fab and Fc regions that have been engineered to generate single unpaired cysteine residues for site-specific conjugation using the THIOMABTM approach.18-23 However, the engineering and conjugation of multiple unpaired cysteines in the antibody Fc region has not yet been comprehensively investigated, and there are limited reports related to the introduction of double or triple cysteines…’    

Response: The suggestion has been accepted and the sentences have been revised as suggested (highlight in blue).

2)  The rationale behind the combination of novel mutations with A118C is not sufficiently well described, nor is the original literature on A118C cited at this point. Please rectify.

Response: The change has been made with an additional sentence for rationale “These top single cysteine mutations were also combined with A118C from CH1 region, which has been shown to generate site-specific ADCs with increased therapeutic index.16” (highlight in blue).

3)  The word ‘good’ in the phrase ‘provide a good case study’ is superfluous

Response: The word ‘good’ in the phrase has been deleted (highlight in yellow/strikethrough).

Results:

1)  Figures 2, 6, S1, S2, S3 and S4 – please indicate the molecular weights of the standards as you have done in Figure S5

Response: They have been included in Figures 2, 6, S1, S2, S3, but not S4 that is western blot result. Although we ran the prestained protein ladder which was transferred into PVDF and we know approximate band migrations, the protein ladder didn’t react with chemiluminescence substrate and didn’t show up in the western blot.

2)  Please include a brief rationale for the use of PDB 1E4K to identify suitable mutation sites. There are many more recent, and significantly higher resolution structures (e.g. 5jii amongst others) that could provide a more precise starting point for your analysis.

Response: The brief rationale has been included “The structure (PDB 1E4K) of IgG1 Fc in complexed with FcγRIII was used for identifying sites for conjugation with minimal impact on FcγR interaction although the receptor is not shown in the figure”(please see highlight in blue).

3)  The words residues, amino acids and side chains in the sentence listing the chosen mutation sites and giving their character (polar, charged, nonpolar) are unnecessary.

Response: They have been deleted.

4)  Table 1 – nanoDSF measurements can also give an indication of aggregation – did these measurements agree with the non-reducing SDS-PAGE analysis? 

Response: This is good suggestion, but unfortunately, we didn’t measure aggregation using nanoDSF at that time.

5)  Table 1 notes – please re-format ml as mL, degrees as ºC, and provide a key to the symbols + and ++

Response: The changes have been made according to the comments from the reviewer.

6)  Last sentence before Figure 2: do you mean S298C not N298C?

Response: Yes, we fixed the typo. Thanks for letting us know.

7)  Figure 2 - The hyperglycosylated mutant A118N NNAS and the A118N mutant are not introduced or set in context – please amend the introduction and/or results section to explain why these mutants have been included. Please define the abbreviation NNAS

Response: The hyperglycosylated mutant A118N NNAS and the A118N mutant were used only as controls for the SDS-PAGE. So, they don’t need to be mentioned in the introduction and/or results section. However, we explained and defined them in the figure legend (see highlight in blue).

8)  Figure 2 – please label the molecular weight standards lane with MW Std as specified in the legend.

Response: The standard lane has been labeled as suggested.

9)  Figure 2 – left hand gels – I think there is a mistake in the labelling – N297Cn should be N297C?  Correct.

Response: Yes, they are typos, and it has been fixed. Thanks!

10)  Figure 2 – where possible, please apply the same labelling as used on Figure 6 to indicate the identity of the different species.

Response: They have been included with labels for non-reducing SDS-PAGE on the left side of gel and one on the right side of gels under reducing condition.

11)  In the phrase ‘Some other mutants showed high PAR…’, please be more specific and replace ‘some’ with a number.

Response: They have been changed according to the comment (see highlight in blue).

12)  Figures 3 and 7 – please indicate the cut offs that you applied with a horizontal line on these bar graphs, and note this in the figure legend.

Response: The cut offs have been applied in all the panels from both Figures 3 and 7, and they were noted in the figure legends (highlight in blue).

13)  Figure 4 legend – please correct the legend to read The ‘percent of mono-PEG at 0 hr’ represents the band area of mono-PEGylated species in the sample at 96 hours divided by that at time 0, and then multiplied by 100.

Response: The sentence has been corrected as suggested (highlight in blue). Thanks for the input.

14)  The sentence ‘Four single cysteine mutants, A339C, K274C, G385C and S440C, showed better conjugation efficiency and selectivity when either reducing agents were used.’ Lacks clarity. I believe what you mean is ‘Four single cysteine mutants, A339C, K274C, G385C and S440C, showed good conjugation efficiency and selectivity regardless of which reducing agent was used.’? 

Response: We appreciate the suggestion from the reviewer, and the sentence has been revised (highlight in blue).

15)  Figure 5 – the words PAR and selectivity in the key are superfluous since they are already in the y-axis title.

Response: The words PAR and selectivity in the key have been deleted.

16)  Figure 5B: please move the x-axis labels to the bottom of the graph below all the bars, so that they can be read. 

Response: The x-axis in the panel has been moved to the bottom of the graph as suggested.

17)  The section titled Expression and characterization of engineered double cysteine mutants lacks clarity and conciseness. I suggest rewording thus: Thirty-eight double cysteine mutants were designed based on the results of PEGylation screening of single cysteine mutants. The top ten single cysteine mutations from PEGylation with DTT reduction (with the exception of Q418C) were combined with each other or with the previously reported A118C.16 They were generated using site-directed mutagenesis and expressed from Expi293 cells. All but one of the engineered double cysteine mutants show comparable expression titers. The exception is the double cysteine mutant K360C+K290C, which shows at least 4-fold reduced titer (Table 2). SDS-PAGE showed high levels of aggregation for the mutant S440C+N384C (Figure S5), while SEC-HPLC analysis also detected high levels of aggregates for this double cysteine mutant, as well as A339C+S440C. The thermal stability of these mutants was also investigated. For six of the seven A339C-containing double cysteine mutants for which Tm1 could be measured, this was reduced significantly (ΔTm1 > -2 ºC) compared to wild-type antibody (Table 2). 

Response: The paragraph has been revised according to suggestion from the reviewer. Thanks.

18)  Table 2 – please define NA in the notes, and reformat degrees as ºC. 

Response: We have replaced NA with ND and defined the word in the notes. The degree was replaced with ºC as suggested (highlight in blue in the notes.

19)  The final list of ‘top’ double mutants might be easier to follow if the amino acids were given in order: i.e. A118C paired with x, y, z; K274C paired with x, y, z; K290C paired with x, y, z and A339C paired with x, y, z. Although I appreciate this will then deviate from how you list the amino acids in Table 2. What was the rationale for arranging in the order that you have chosen for Table 2 (and 3, and Figure 8), rather than amino acid residue number? 

Response: We don’t have specific rationale for the arrangement. So, the text (highlight in blue) as well as Figure 8 as well as tables 2 and 3 have been revised with the amino acid residue number according to the suggestion. We hope that they are better now.

20)  Figure 6. What do you mean by ‘the original mutated clone with the correct sequence’? Do you mean here simply the correct, expected clone A118C + A339C? Perhaps this would be clearer if you were to re-label the incorrect clone with its actual sequence, whatever that turned out to be?

Response: No, the first clone chosen for expression have a sequence error. So, we have chosen a second clone with correct sequence for expression. The legend has been revised to have thing clarified. Sorry for the confusion.

21)  Figure 7 – the text on the x- and y-axes is completely illegible as it is too small – please fix this.

Response: We have increased the size of the text and hope that they are okay now. Thanks for letting us know.

22)  Figure 8 – the text on the figure is too small to be legible – please fix this.  

Response: The size of the text on the figure has been increased, and we hope that they are okay now.

23)  Figure 8 – the definitions for the colouring scheme in the figure legend and the text to the right of the figure do not agree – which is correct? Please remove the incorrect text. The figure legend should be sufficient

Response: The incorrect one has been deleted, and only figure legend is shown. Thanks.

24)  Please consider re-phrasing the last paragraph of the results along these lines: ‘The top double cysteine mutants demonstrated not only high coupling efficiency but also good selectivity. The A118C, A339C, and K274C mutations are highly compatible with other cysteine mutants including each other. The PEGylation of large numbers of cysteine-containing mutants allows a simple and straightforward screening to identify antibodies with PAR over 3.4 and with minimal off-target conjugation.’ 

Response: We appreciate the suggestion from the reviewer, and the paragraph has been revised based on the input.

Discussion:

1)  Rationals should be rationale

Response: The typo has been fixed (highlight in blue).

2)  Please specify the number of cysteine residues rather than stating ‘we engineered many single and double cysteine residues’ and later ‘Many double cysteine mutants have been identified’

Response: The changes have been made according to the comment (highlight in blue).

3)  Consider replacing ‘more’ with ‘additional’ or even ‘x additional sites’ (where x = the number of novel/additional sites)

Response: We have revised the text as suggested (highlight in blue).

4)  In your list of mutants that were evaluated in earlier work, you include S239C and S442C, however, these are not mentioned in the preceding sentence to which this sentence refers?

Response: The sentence should refer the mutations evaluated in early work. To clarify this, the sentence has been revised (highlight in blue).

5)  In the sentence ‘selectivity is also a relevant specification’, I think parameter may be a better word to use.

Response: The sentence has been changed based on the comment (see addition highlighted in blue and deletion in yellow/strikethrough).

6)  Instead of ‘less than expected’, use ‘incomplete’ for conciseness.

Response: The word has been replaced as suggested (addition highlighted in blue and deletion in yellow/strikethrough).

7)  Instead of ‘always be good to’ use ‘be useful/beneficial/helpful to’

Response: The phrase ‘always be good’ has been replaced with ‘be beneficial’ (addition highlighted in blue and deletion in yellow/strikethrough).

Methods:

  • Please state which plasmid/vector(s) were used for the mutagenesis and expression or provide a citation for prior work. As currently worded, the mutagenesis and transfection sections are not fully reproducible in another lab.

Response: The whole paragraph has been rewritten. We have included plasmid information, and more details on mutagenesis and transfection have been provided included the kits used as well as catalog numbers (highlight in blue).

  • What do you mean by ‘96-well plate format with 0.5 mL culture media at 3- or 5-mL scale’? What was 0.5 mL and what was 3/5 mL? What cell density/cell count did you use? And what method and reagents were used to do the transfections?

Response: We are sorry for the confusion, and the paragraph has also been rewritten with more information. To clarify the procedure used, a couple of sentences were added “The cells were transiently transfected with mutated DNA plasmid using Expi293 expression system (ThermoFisher Scientific, Catalog No. A14635) in 96-well plates at 1.25 x 106 cells/0.5 mL/well, according to the manufacturer's instructions. Each mutant was expressed in multiple wells (6 or 10 wells) to harvest 3 or 5 mL of conditioned media for Protein A purification.” (see highlight in blue). We hope that it is clear now and we have answered the questions from the reviewer.

3)  What recipe/make/supplier of loading buffer did you use for PAGE? And what make/supplier of gels? 

Response: The suppliers of loading buffer and gels have been added (highlight in blue).

4)  Please make sure to provide the reference for the AlphaView software throughout the text (i.e. in the figure legends also). 

Response: The reference has been included as suggested (highlight in blue).

5)  Please provide either more details on the PEG staining (time, temperature) or a citation.

Response: A citation has been added (highlight in blue).

6)  CO2 should be CO2  

Response: Change has been made (highlight in blue).

7)  Please provide details of the supplier etc for the PVDF and details of the western transfer process (wet, semi-dry, fast?) 

Response: The information has been provided (highlight in blue).

8)  Please provide catalogue numbers for all antibodies 

Response: The catalogue numbers have been provided (highlight in blue).

9)  Please provide details of the instrument, capillaries, buffers and protein concentrations used for the nanoDSF measurements.

Response: The whole paragraph has been rewritten with detail information related to the instrument, capillaries, buffers, and protein concentrations being provided according to the suggestion.

Supplementary information:

1) Did you intend for your supplementary to include the main figures as well?  

Response: No, they are just supplementary figures. We will change them to a separated file if possible.

2) Supplementary figure legend S5: SeaBlue Plus 2 should be SeeBlue Plus2   

Response: We appreciate for letting us know the typo. It has been fixed.

3) Note that in the second instance in the Results where you refer to Figure S4, this should be Figure S5. 

Response: The mistake has been corrected. Thanks.

Round 2

Reviewer 1 Report

Zhou et al. submitted an improved version of their manuscript for re-evaluation, based on concerns raised in the previous round of revision. Since no additional concerns are raised, I recommend the manuscript for publication.

Author Response

We thank the reviewer very much for all his or her comments and helps!

Reviewer 2 Report

Dear Dr. Zhou and co-authors,

Thank you for your responses to my suggestions. I would like to suggest some additional minor amendments to the revised version before the article is published in order to improve the clarity of the language used, and to fully incorporate the suggested changes to the figures, which are described in the legends but seem to be missing from the figures.

Please consider rewording the following sentence in the section on the conservation of the mutations across IgG subclasses from “Even with the rest of six sites, four residues are identical among IgG1 and IgG4 which are mostly used for monoclonal antibody therapy, suggesting the generality of these twenty-seven sites for potential antibody conjugation” to “Of the six sites that are not fully conserved, four residues are identical between IgG1 and IgG4, which are the most commonly used for monoclonal antibody therapy, suggesting the generality of these twenty-seven sites for potential antibody conjugation.”

Also, I note that the revised colour scheme described in the legend to Figure 1 does not match the figure? (legend says yellow for C229, figure shows red? CH2 sites are described as in magenta, figure shows orange/coral?

Figure 2 - the molecular weights (in kDa) and the species labels are missing from the figure although described in the legend?

Figure 3 - the figure legend describes cutoff lines and top ten arrows, but these are missing? the reformatting of the axes and addition of panel labels has not been done.

Figure 5 - the reformatting of the axes and series labels has not been done. The arrows described in the figure legend are missing from the figure.

Figure 6 legend - the description of the two set of results for A118C + A339C is a little clearer now, but could be further improved as follows: "The initial clone for mutant, A118C+A339C (labelled in red), showed poor expression and diffuse bands after PEGylation. It was re-sequenced and found to have a sequence mismatch. A second clone clone for A118C + A339C with the correct sequence was expressed and PEGylated as shown (labelled in black)."

Figure 7; cutoffs, arrows and panel labels are missing from the figure, although described in the legend. Font size has not been increased. Axis formatting has not been changed.

Figure 8 still contains the mismatched description of the ranges in the legend vs. the figure itself and the font size has not been increased.

Line 303; I think you may mean "limited both in their..." rather than 'nor?

Line 463 - do you mean that wild-type antibody is included in both (rather than either) conjugation and SDS-PAGE?

Line 528: I think you mean PEGylation? or % PEGylated rather than simply PEGylated?

Line 475 remove the full stop after the word hours.

Line 407; 'others five' should be either 'another five' or 'five others'

Lines 562-563 - the definitions of + and ++ are still unclear to this reader. Do you perhaps mean that + indicates the presence of a band for half-antibody or aggregate, whilst ++ indicates that these samples contain the strongest intensity bands for half-antibody/aggregate of all analysed samples?

With regard to the lack of molecular weight indication on Figure S4, I appreciate that the standards are not detectable with chemiluminescence, however, it is best practise to indicate their location on the film (if used), or via orthogonal detection in the colorimetric channel (if using a scanner/imager). If you know where one or more standard(s) runs, you should still indicate the position(s) on the western with a dash or arrow, and give the molecular weight(s) to allow the user to orient themselves.

Best wishes for your future research.

Author Response

Dear Dr. Zhou and co-authors,

Thank you for your responses to my suggestions. I would like to suggest some additional minor amendments to the revised version before the article is published in order to improve the clarity of the language used, and to fully incorporate the suggested changes to the figures, which are described in the legends but seem to be missing from the figures.

Response: The authors appreciate the excellent comments and time the reviewer spent for improving our manuscript. We actually spent significant amounts of time in revising the figures and tables based on inputs from the reviewers. However for some reason, our updated version of figures wasn’t applied to this revised manuscript in first round reviewing. It seems that the reviewer still looked at our old version of figures which are still used and formatted here for 2nd round of reviewing. We appreciate the reviewer to point them out, and the old figures have been replaced in the new updated version.

  • Please consider rewording the following sentence in the section on the conservation of the mutations across IgG subclasses from “Even with the rest of six sites, four residues are identical among IgG1 and IgG4 which are mostly used for monoclonal antibody therapy, suggesting the generality of these twenty-seven sites for potential antibody conjugation” to “Of the six sites that are not fully conserved, four residues are identical between IgG1 and IgG4, which are the most commonly used for monoclonal antibody therapy, suggesting the generality of these twenty-seven sites for potential antibody conjugation.”

Response: The sentence has been revised according to the comment from the reviewer.

  • Also, I note that the revised colour scheme described in the legend to Figure 1 does not match the figure? (legend says yellow for C229, figure shows red? CH2 sites are described as in magenta, figure shows orange/coral?

Response: As mentioned above, Figure 1 formatted here is our old version of figure but not updated one.

  • Figure 2 - the molecular weights (in kDa) and the species labels are missing from the figure although described in the legend?

Response: Same as above.

  • Figure 3 - the figure legend describes cutoff lines and top ten arrows, but these are missing? the reformatting of the axes and addition of panel labels has not been done.

Response: Same as above.

  • Figure 5 - the reformatting of the axes and series labels has not been done. The arrows described in the figure legend are missing from the figure.

Response: Same as above.

  • Figure 6 legend - the description of the two set of results for A118C + A339C is a little clearer now, but could be further improved as follows: "The initial clone for mutant, A118C+A339C (labelled in red), showed poor expression and diffuse bands after PEGylation. It was re-sequenced and found to have a sequence mismatch. A second clone clone for A118C + A339C with the correct sequence was expressed and PEGylated as shown (labelled in black)."

Response: The change has been made and we appreciate the suggestion from the reviewer.

  • Figure 7; cutoffs, arrows and panel labels are missing from the figure, although described in the legend. Font size has not been increased. Axis formatting has not been changed.

Response: Same as above.

  • Figure 8 still contains the mismatched description of the ranges in the legend vs. the figure itself and the font size has not been increased.

Response: Same as above.

  • Line 303; I think you may mean "limited both in their..." rather than 'nor?

Response: No, there is not much examples which show site-specific antibody conjugation generating ADC with a DAR greater than two.

  • Line 463 - do you mean that wild-type antibody is included in both (rather than either) conjugation and SDS-PAGE?

Response: No, we sometimes included the wild-type antibody in conjugation to validate the procedure, and of most of times just include it in SDS-PAGE run since there are so many single and double mutants for screening. I hope that it makes sense.

  • Line 528: I think you mean PEGylation? or % PEGylated rather than simply PEGylated?

Response: Yes, you are right. The word has been replaced with “PEGylation”. Thanks.

  • Line 475 remove the full stop after the word hours.

Response: The typo has been deleted.

  • Line 407; 'others five' should be either 'another five' or 'five others'

Response: We appreciate the suggestion and the words have been changed.

  • Lines 562-563 - the definitions of + and ++ are still unclear to this reader. Do you perhaps mean that + indicates the presence of a band for half-antibody or aggregate, whilst ++ indicates that these samples contain the strongest intensity bands for half-antibody/aggregate of all analysed samples?

Response: Yes, we rewrote the definition as suggested.  

  • With regard to the lack of molecular weight indication on Figure S4, I appreciate that the standards are not detectable with chemiluminescence, however, it is best practise to indicate their location on the film (if used), or via orthogonal detection in the colorimetric channel (if using a scanner/imager). If you know where one or more standard(s) runs, you should still indicate the position(s) on the western with a dash or arrow, and give the molecular weight(s) to allow the user to orient themselves.

Response: Figure S4 has been revised according to the suggestion from the reviewer. 

  • Best wishes for your future research.

Response: Thank you so much for the help and kindness!